# Practical Bayesian Guidelines for Small Randomized Oncology Trials

**DOI:** 10.3390/cancers17121902

**Published:** 2025-06-07

**Authors:** Peter F. Thall

**Affiliations:** Department of Biostatistics, MD Anderson Cancer Center, Houston, TX 77030, USA; rex@mdanderson.org

**Keywords:** Bayesian statistics, clinical trial, feasibility, randomization, safety monitoring, futility monitoring

## Abstract

Randomization is used infrequently in small early-phase clinical trials of several different treatments or multiple doses of a single agent. When a trial’s goals include comparing treatments or doses based on early clinical outcomes, however, randomization provides much more useful data than single-arm trials because it facilitates fair between-treatment comparisons. Randomization does this by preventing confounding of treatment effects with between-study differences in the distributions of prognostic variables. This paper provides Bayesian criteria for estimating treatment effects to facilitate the planning and analysis of small randomized trials. Practical guidelines are given for determining sample sizes, choosing the number of treatment arms, specifying safety and futility monitoring rules, and constructing a balanced randomization scheme. The methods are illustrated by a trial of engineered cells to treat steroid-refractory graft-versus-host disease.

## 1. Introduction

Although randomization is not commonly used in small early-phase trials, recognition by clinical investigators that such trials are inherently comparative has led increasingly to its use to obtain fair treatment comparisons [1,2]. The question of whether to randomize patients in early-phase treatment evaluation has a long history, and it remains controversial [3,4,5,6,7,8,9,10,11]. In this paper, we argue that randomization is very useful in small, early-phase clinical trials rather than only in large trials. Our primary goals are to convince clinical trialists to randomize in small trials and to describe practical Bayesian methods for planning and analysis of small randomized clinical trials (SCRTs), including making comparative inferences from small samples.

SRCTs of two or more treatments or of multiple doses of a new agent, play a key role in the treatment evaluation process. Data about treatment feasibility, safety, and early efficacy in humans may be used either as a bridge between preclinical experiments and a large confirmatory phase 3 trial or to decide that a phase 3 trial is not warranted. In practice, overall sample sizes of early-phase trials are determined primarily by resource constraints, including financial costs, accrual rate, and availability of the new treatment or treatments being studied. Consequently, rather than presenting formulas for computing sample sizes using power calculations based on tests of hypotheses, we provide a heuristic approach to sample size determination. This includes assessing both practical constraints and the statistical reliability of per-arm sample sizes in terms of Bayesian posterior credible intervals for estimating between-treatment effects.

An SRCT with K = 2, 3, or 4 treatment arms and N = 20 to 60 patients may be conducted to choose the best dose, schedule, or treatment, screen out unsafe or ineffective treatments, or obtain preliminary treatment comparisons. Examples include trials to make preliminary comparisons of different engineering processes of cellular immunotherapy for hematologic malignancies, optimize the dose of a targeted molecule, or evaluate a biologically targeted agent. Most commonly, the clinical effects of a new treatment are characterized by the probabilities of early clinical response (Res), severe toxicity (Tox), and possibly biological variables related to treatment. Given longer follow-up, mean or median progression-free survival (PFS) time or overall survival (OS) time may also be estimated. With randomization, preliminary estimates of PFS or OS time distributions for the experimental treatments being studied and standard of care may provide an empirical basis for making a “Go–No Go” decision of whether to conduct a phase 3 trial.

An SRCT that uses Res and Tox for treatment evaluation and interim safety or futility monitoring may be regarded as a randomized phase 2 or phase 1–2 trial [5,12,13,14]. An SRCT provides a scientifically attractive alternative to conducting a single-arm phase 1 trial based on Tox alone followed by an expansion cohort or a single-arm phase 2 trial based on Res. This is in accordance with FDA Project Optimus [15,16], which was initiated to address the problem that many doses chosen in conventional phase 1 trials later are found to be excessively toxic or ineffective in a phase 3 trial or clinical practice, leading to ad hoc dose adjustments. A key recommendation of Project Optimus was to randomize patients among doses when appropriate. Provided that safety monitoring rules to stop accrual to overly toxic doses are included, and if there is no compelling reason to assume that either Pr(Tox) or Pr(Res) must increase with dose, randomization is ethical. Otherwise, a sequentially adaptive dose-finding method may be more appropriate than randomizing patients among doses [13].

It is well established that conventional “3 + 3” algorithms used for dose finding in many phase 1 trials are likely to make bad decisions when choosing a maximum tolerable dose (MTD) or a recommended phase 2 dose (RP2D) [13,14,17,18]. While there are many different 3 + 3 algorithms, nearly all choose doses sequentially for successive cohorts of size 3, starting the trial with the first dose level, which often is either the lowest dose or the next to lowest dose, specified by the investigators prior to the trial. For a new dose where patients have not yet been treated, if no DLT is seen in any of the first cohorts of 3 patients, denoted by 0/3, then 3 additional patients are treated at the next higher dose level. If, instead, 1/3 of patients have a DLT at a new dose, then 3 more patients are treated at that dose. Dose escalation continues until 2 or more patients, out of either 3 or 6 patients treated at a dose, experience DLTs, that is, if 33% or more patients have a DLT at a new dose level. In this case, the dose is considered excessively toxic, and the MTD is defined to be one dose level below the excessively toxic dose. A variant of this algorithm requires that at least 6 patients must be treated at the MTD to obtain better reliability. However, this rule may lead to a problem; for example, an initially chosen MTD later may turn out to have 2 or 3 DLTs in 6 patients, in which case further de-escalation is needed. All 3 + 3 algorithms carry a high risk that the selected MTD or RP2D will cause severe toxicity, primarily because the per-dose sample sizes are far too small to estimate Pr(DLT) reliably.

For example, a phase 1 trial of the tyrosine kinase inhibitor ponatinib for treating Philadelphia chromosome-positive leukemias using a 3 + 3 algorithm chose the unsafe dose of 45 mg PO daily [14,19]. This later was modified to start with 45 mg PO daily but drop to 15 mg PO daily once ≤1% BCR-ABL was achieved. A phase 1 trial of the monoclonal antibody onartuzumab for treating non–small cell lung cancer using a 3 + 3 algorithm chose a dose with a low response rate [14,20]. This might have been avoided, for example, if a phase 1–2 design using both Res and Tox had been used [12,13,14].

In general, any phase 1 dose-finding design based on Tox alone has a substantial risk of choosing an ineffective dose because it ignores Res. For example, in the extreme case where no responses are seen at any dose, a conventional phase 1 design still will choose an MTD or RP2D, despite the fact that the observed data show the selected dose is likely to be completely ineffective. This problem may also arise with the continual reassessment method (CRM) [17], which chooses each new cohort’s dose to have an estimated Pr(Tox) closest to a fixed target probability. This relies on the underlying assumptions, which are seldomly stated, that both Pr(Tox) and Pr(Res) increase with dose and that there is a trade-off between the risk of Tox and the chance of Res at any given dose. These assumptions were originally motivated by studies of cytotoxic agents, but they may not hold for biological agents, such as cellular therapies. For example, a CRM design with a Pr(Tox) target of 0.25 considers a dose with Pr(Tox) = 0.40 more desirable than a dose with Pr(Tox) = 0.05. A possible rationale for this is the belief that the dose with Pr(Tox) = 0.40 is likely to have a higher Pr(Res) than the dose with Pr(Tox) = 0.05 and that this unknown higher response rate is a trade-off for the 40% Tox rate. This example illustrates why any early-phase clinical trial design should include explicit decision criteria that use both Res and Tox, rather than only using Tox [12,13].

The goals of this paper are to explain problems that arise from conducting early-phase trials without randomizing, and to provide practical Bayesian methods for design, conduct, and analysis of SRCTs. It will be assumed that the goals of an SRCT are to obtain preliminary comparative estimates of key parameters, including Pr(Res), Pr(Tox), and possibly mean or median PFS and OS time, and to use the comparative estimates as a basis for deciding how to proceed in the treatment evaluation process. Practical guidelines are provided for computing and interpreting Bayesian posterior estimates, planning sample sizes, constructing Bayesian safety and futility monitoring rules, and specifying a balanced randomization scheme. The methods are illustrated by a real-world oncology trial, and the paper closes with a brief discussion.

## 2. Confounding, Bias, and Randomization

In all that follows, the usual statistical distinction will be made between a parameter, which is a property of a patient population being studied, such as Pr(Res) or Pr(Tox) for a given treatment or dose, and a statistical estimator of the parameter computed from data. For a two-arm SRCT comparing an experimental treatment, E, to the standard of care, S, for brevity, denote θ_k_ = Pr(Res with treatment k) for k = E or S. The trial’s data may be used to obtain a preliminary estimate of θ_E_ − θ_S,_ the comparative E-versus-S effect on the response rate, and similar parameters for TOX and PFS. Similarly, for a three-arm SRCT of S and two experimental treatments, E_1_ and E_2_, the comparative experimental-versus-S effects are θ_E1_ − θ_S_ and θ_E2_ − θ_S_. While the precision of comparative effect estimators is limited by the small per-arm sample sizes of an SRCT, the estimators are still useful as an empirical basis for deciding how to proceed in the clinical evaluation process. Possible decisions may be to discard one or more experimental treatments due to excessive toxicity or ineffectiveness or to investigate one or more of the treatments further in a larger randomized trial based on long-term PFS or OS. The ultimate goal of a sequence of clinical trials is to provide an empirical basis for deciding whether to replace S with a new treatment in clinical practice.

The causal motivation for randomization may be explained by the following thought experiment [21]. Suppose that one could make two identical copies of each patient, treat one copy with E and the other with S, and observe their future potential outcomes, Y_(E)_ and Y_(S)_. The difference, Y_(E)_ − Y_(S),_ then would be the *causal effect* of E versus S on the patient. Repeating this for all patients in a trial and averaging would provide a sample mean causal effect. While this experiment is impossible and a causal effect cannot be observed [21], it provides a conceptual basis for proving mathematically that randomization gives an unbiased statistical estimator of the mean causal effect. That is, if patients are randomized, then the approximate mean of a conventional estimator is θ_E_ − θ_S_ [22,23].

To see the advantages of randomizing patients when comparing treatments, it is useful to consider what can go wrong if one does not randomize. Suppose that separate trials of E and S are conducted or a single-arm trial of E is conducted with the plan to use historical data on S for comparison. The actual estimand of a conventional statistical estimator based on data from such trials is θ_E_ − θ_S_ + [between-trial effect], rather than the E-versus-S effect θ_E_ − θ_S_. That is, a conventional estimator is *biased* because the between-treatment effect of interest is confounded with a between-trial effect that arises from systematic differences in patient characteristics between the E and S datasets. The result is that an apparent treatment effect difference obtained using a conventional statistical estimator computed from non-randomized data may be due, in part or entirely, to the effects of patient prognostic covariates that are unbalanced between the two datasets. A numerical example of this problem is given in Table 1a, which shows true response probabilities and data for E and S in Good and Poor prognosis patient subgroups. While the true response probabilities are assumed to be known in this example, in practice, they are not known and must be estimated from available data. Suppose that a single-arm trial of E enrolls only good prognosis patients and has a sample response rate of 15/30 (50%), while historical data on S, including both good and poor prognosis patients, give overall response rate 36/100 (36%). If prognosis is ignored, or if this unfair sampling is not known, then it appears that E has a higher overall response rate than S. If, instead, one knows that the E patients all had good prognosis while the S sample included both good and poor prognosis patients, then it is obvious that the comparison is unfair. While this example is very simple, in practice, non-comparable samples may arise in numerous ways from non-randomized trials, and in many settings, between-study bias may be due to external variables that are not known.

A common misconception is that if one wishes to compare E to S, conducting a small single-arm trial of E, often with 20 to 40 patients, is perfectly acceptable because historical data on S may be used for comparison. This is based on the mistaken belief that one can correct for confounding, for example, by fitting a conventional logistic regression model for Pr(Res) or a survival time regression model such as a Cox model for PFS, to the combined data, if the model includes key prognostic variables [24]. It is well known that, in general, this is not true [25]. A naïve regression analysis of non-randomized data on E and S may easily give biased estimators of between-treatment effects.

Well-established methods to correct for bias when analyzing non-randomized, observational data include inverse probability of treatment weighting (IPTW), pair matching, and generalized estimation [26,27,28,29,30,31,32]. IPTW uses the *propensity score* of each patient, which is a statistical estimate, p*(**Z**), based on the patient’s baseline covariates, **Z**, such as age, disease severity, or other characteristics, of the probability p(**Z**) that they would receive the treatment that they actually received. The estimate p*(**Z**) may be obtained by fitting a logistic or probit model for the patient treatment indicator as a function of **Z**. For IPTW estimation, each patient’s outcome Y, which, for example, may be an indicator of Res, or possibly PFS time, is replaced with the weighted value Y/p*(**Z**). The aim is to correct for the possible biasing effect that **Z** may have had if it was used to choose patients’ treatments. A simpler method, which often works surprisingly well in practice, is to include p*(**Z**) as an additional covariate in a regression model for Y as a function of **Z**. These bias correction methods are practical if the trial and historical samples are both sufficiently large and both samples include key patient covariates related to the clinical outcomes being compared. In practice, however, sample sizes of single-arm trials often are too small to implement bias correction methods reliably, and moreover, some key covariates may not be available for all patients in the trial and historical datasets. A common practice when reporting the results of a single-arm trial is to give estimated rates of Res, Tox, and other outcomes while citing corresponding historical rates seen with S. This implicitly invites the reader to compare numerical values of estimators that are not comparable due to confounding by between-study effects. Similarly, for PFS or OS time, plotting two Kaplan-Meier curves based on data from separate trials of E and S is a common example of this practice since it leads the reader to visually compare survival curves that are not comparable. The practical consequence of failure to randomize in a small trial is that it is likely to produce data that are of little use for comparing treatments fairly and are misleading due to confounding [33].

If E is a biologically targeted agent, there may be important additional issues to address in an SRCT. The phrase “precision (personalized, individualized) medicine” is often used to refer to the use by physicians of biomarkers that designed molecules or immunological agents have been engineered to attack to choose each patient’s treatment [34,35,36,37,38]. Based on its construction, a biological agent should have a greater anti-disease effect than S in patients who are biomarker-positive. For example, vascular endothelial growth factor (VEGF) inhibitors, such as bevacizumab, sorafenib, and sunitinib, are targeted agents designed to reduce blood flow to a tumor by blocking its angiogenesis. For such agents, the biomarker indicates that the patient is VEGF positive. Table 1b illustrates a setting where a targeted agent, E, is highly effective in biomarker-positive patients, with a true Res rate of 80%, while the Res rate of E drops to 10% for biomarker-negative patients. The Res rate of S is 50% regardless of biomarker status. In this setting, preclinical in vitro or in vivo data may suggest that comparing E to S in humans is relevant only in biomarker-positive patients. In this setting, averaging the response rates of E for biomarker positive and negative patients to obtain one overall rate makes little sense since the optimal treatment may not be the same in these two subgroups. A fair apples-to-apples comparison for biomarker-positive patients correctly shows that E is greatly superior to S in that subgroup, where patients have the biological target that E is engineered to attack. In such settings, it may make sense to conduct an SRCT of E versus S in biomarker-positive patients only. However, an important *caveat* is that E also may have a meaningful anti-disease effect in biomarker-negative patients due to an undiscovered biological pathway. Thus, because there often is much to be learned about a new targeted agent in an early-phase trial, it may be worthwhile to enroll biomarker-negative as well as positive patients. In any case, E-versus-S effects should be estimated separately in the biomarker positive and negative subgroups.

## 3. Bayesian Inference

Bayesian methods are particularly well-suited for making inferences from small samples, constructing practical safety and futility monitoring rules for clinical trials [38,39], and making predictions [40,41,42,43]. Because Bayesian inferences are valid for any sample size, they avoid the problem that many frequentist methods, such as estimators of treatment effects obtained from fitted Cox or logistic regression models, rely on asymptotic statistical distribution theory that is not valid for small samples.

A Bayesian model includes two types of objects. The first is observable variables, such as indicators of Res or Tox, or numerical values of PFS, OS, or last follow-up time. The second is parameters, denoted by θ, which are conceptual quantities such as Pr(Res), Pr(Tox), or median PFS time with a given treatment. The Bayesian paradigm considers θ to be random and includes a prior distribution on θ. The randomness of data is characterized by a likelihood function, such as a binomial distribution for count data or an exponential distribution for event times. Bayes’ theorem combines the prior with the likelihood of the observed data to obtain a posterior, p(θ|data), which is used to make inferences about θ. A common Bayesian estimator is the posterior mean, which is a weighted average of the prior mean and the sample mean. To quantify uncertainty about θ, it is useful to accompany the posterior mean by a 95% posterior credible interval (CrI), which by definition is a pair of numbers [L, U] for which Pr(L < θ < U|data) = 0.95. For example, to represent little prior knowledge, it may be assumed that θ = Pr(Res) follows a beta(0.50, 0.50) prior, which has a mean of 0.50 and effective sample size ESS = 0.50 + 0.50 = 1. For binomial data consisting of X = number of responses out of n patients with θ = Pr(Res), the posterior p(θ|X) is beta(0.5 + X, 0.5 + n − X). For example, if X = 8 responses are observed in n = 20 patients, then θ follows a beta(8.5, 12.5) posterior, which has a mean 8.5/21 = 0.405 and gives 95% CrI [0.21, 0.62] for θ. More generally, if θ follows a beta(a, b) prior, which has mean a/(a + b), then the posterior p(θ|X, n) is beta(a + X, b + n − X), which has mean (X + a)/(n + a + b), and may be written as the weighted average {n/(n + a + b)}(X/n) + (a + b)/(n + a + b) {a/(a + b)}. This gives weight n/(n + a + b) to the sample proportion X/n, and weight (a + b)/(n + a + b) to the prior mean a/(a + b), thus “shrinking” the conventional estimator X/n toward the prior mean a/(a + b). All Bayesian estimators have this shrinkage property, which provides more stable estimators and reduces the effects of sampling errors and the risk of overfitting data. Median PFS time may be estimated similarly by assuming an exponential-gamma Bayesian model.

There is extensive literature on how a prior should be specified for a Bayesian model [40,41,42,43]. A strict Bayesian analysis requires a prior to be elicited from one or more area experts. A common criticism of Bayesian statistics is that an elicited expert prior that is highly informative may lead to inferences that are based mainly on subjective opinions rather than data. For example, suppose that an investigator optimistically believes that the mean of Pr(Res, E_k_) for a new treatment E_k_ is 0.80 and has very little uncertainty so that the investigator’s prior is beta(80, 20). Since this prior has an effective sample size ESS = 80 + 20 = 100, it will dominate any inferences based on a sample of n = 20 patients. As an extreme example, if no responses were observed, the posterior of Pr(Res, E_k_) would be beta(80, 40), which has a mean of 0.67. In contrast, a frequentist analysis has no prior, and for this dataset, it would estimate Pr(Res, E_k_) more simply by using the empirical rate 0/20 = 0. While the beta(80, 20) prior, which arguably is overly informative, leads to a posterior mean estimate that sharply disagrees with the observed data, the frequentist estimate of 0 says that Res is impossible.

To avoid this sort of problem when using a Bayesian model in practice, an “operational” prior typically is assumed to facilitate computation and obtain a sensible data analysis in the setting at hand. For an SRCT, the prior should be non-informative in that it carries a small amount of information, so posterior inferences are dominated by the observed data rather than by the prior. For example, if a beta(a, b) distribution is assumed for Pr(Res, E_k_) in an SRCT, a typical operational requirement is that the ESS = a + b = 1, or at most 2. This is needed so that, for example, the Bayesian monitoring rules described below will have good operating characteristics. A common practical approach is to elicit the physician’s prior mean, set this to equal the beta mean a/(a + b), set a + b = 1, and solve for a and b. For example, if the elicited mean of Pr(Res, E_k_) is 0.40, then a beta(0.40, 0.60) prior is assumed. In the above example where 0/20 responses were observed, one may assume a beta(0.80, 0.20) prior, which has the optimistically large mean of 0.80 but ESS = 1. This leads to a beta(0.80, 20.2) posterior, which has a mean of 0.04 and 95% CrI [0.00, 0.15]. This CrI says that, given the data, there is a 95% chance that Pr(Res, E_k_) is smaller than 0.15. Since it incorporates uncertainty, this Bayesian analysis may be considered more informative than simply saying that the response rate is estimated to be 0.

Comparing Pr(Res), Pr(Tox), or median PFS between arms based on SRCT data provides a quantitative basis for deciding how to proceed with treatment development. For each experimental treatment, E, in the trial, the between-treatment effect, θ_E_ − θ_S_, may be estimated by a posterior mean and accompanying 95% CrI. Additionally, the posterior probability that E provides at least a δ improvement over S in response probability is Pr(θ_E_ > θ_S_ + δ|data), which may be computed for a meaningfully large improvement, such as δ = 0.15 or 0.20.

An SRCT with n = 10 to 30 patients per arm gives Bayesian estimators of between-treatment effects that are approximately unbiased, but they are imprecise due to the small sample size. Figure 1 illustrates how statistical reliability increases with sample size by giving the posteriors and 95% CrI’s of θ_E_ − θ_S_ for each of four two-arm RCT datasets, each with empirical response rates of 40% for E and 20% for S, assuming that both parameters follow a beta(0.50, 0.50) prior. Since a distribution represents probability by area under its curve, in each plot, the shaded area under the curve between L = the 2.5th percentile of the posterior and U = the 97.5th percentile equals 0.95, so [L, U] is a posterior 95% CrI for θ_E_ − θ_S_. The upper left posterior, obtained from samples of n = 20 patients per arm, gives the widest 95% CrI, [−0.08, 0.45], which has a width of 0.53. The CrI’s become successively narrower as the per-arm sample sizes increase from 20 to 40, 100, and 200. Figure 1 illustrates the key point that, when comparing E to S, it is misleading to cite response rates of 40% and 20% without also giving the sample sizes from which they were computed or a CrI or confidence interval to quantify uncertainty. As noted earlier, if patients were not randomized between E and S, then the posterior distribution would be of θ_E_ − θ_S_ + [confounding effects] rather than of θ_E_ − θ_S_. In this case, this sort of Bayesian computation would be invalid, and its results would be misleading. Thus, to compare treatments, randomization is essential.

It is also useful to compare the distributions of two parameters visually by plotting their posteriors together. Figure 2 shows posteriors of θ_E_ and θ_S_ based on samples of size n = 15 per arm (top row) and n= 20 per arm (bottom row). Given observed response rates 7/15 for E and 3/15 for S (upper left), a 95% CrI for θ_E_ – θ_S_ is [−0.07, 0.55], and Pr(θ_E_ > θ_S_|data) = 0.94 but Pr(θ_E_ > θ_S_ + 0.20|data) = 0.63. Thus, θ_E_ is likely to be larger than θ_S,_ but E is not very likely to provide a 0.20 improvement over S in response probability. Observed response rates 10/15 for E and 3/15 for S (upper right) give 95% CrI [0.12, 0.71] for θ_E_ − θ_S_, with Pr(θ_E_ > θ_S_ + 0.20|data) = 0.93, so here E may be considered promising since it is likely to provide a 0.20 improvement over S in response rate. The bottom row gives similar comparisons, where the observed rates are 8/20 for E versus 4/20 for S (lower left), with 95% CrI [−0.08, 0.45] for θ_E_ − θ_S_ and Pr(θ_E_ > θ_S_ + 0.20|data) = 0.48. The data 12/20 for E versus 4/20 for S (lower right) give 95% CrI [−0.08, 0.45] for θ_E_ − θ_S_ and Pr(θ_E_ > θ_S_ + 0.20|data) = 0.90, so in this case E is promising. Again, without randomization, all of these computations would be invalid due to confounding between-treatment effects with between-trial effects. Similar Bayesian posterior computations may be carried out to compare Pr(Tox) or median PFS times between E and S.

## 4. Trial Design Guidelines

Including S as a treatment arm in an SRCT is highly desirable because it provides unbiased answers to the question of how each experimental treatment E compares to S in terms of their Res and Tox rates. If a given E is not eliminated by preliminary screening when compared to S in terms of the observed early Res and Tox rates, then E may be compared to S in a later phase 3 trial based on PFS or OS time. A single-arm trial of E cannot provide this sort of comparative inference.

### 4.1. Determining Sample Sizes

Conventionally, a design for a large randomized clinical trial (RCT) is based on a test of hypotheses, and its sample size is planned by fixing the test’s overall type I error rate, typically at 0.05 or 0.10, and doing power computations for hypothesized improvements of E over S in terms of median PFS or OS time [44,45]. In contrast, small early-phase trials are conducted to obtain preliminary estimates of Res, Tox, and PFS rates that are used to screen treatments and plan future trials. Thus, rather than being used to test hypotheses, small sample parameter estimates obtained from SRCTs may be used to generate hypotheses for testing in future trials.

For an SRCT, given K = the number of treatments to be studied and n = the per-arm sample size, the maximum total sample size is N = Kn. As noted above, in theory, one might formulate hypotheses in terms of P(Res) to compare each E_k_ to S, specify a statistical test of the hypotheses, and perform a power computation to derive N. In practice, such computations are of limited use because N is determined primarily by financial constraints, drug availability, accrual rate, and K. A typical SRCT has N = 20 to 60 patients and K = 2, 3, or 4 arms. Since N = Kn, a simple heuristic approach for determining (N, K, n) is to examine how a design behaves for a few possible triples. This may be carried out by quantifying how precisely θ_E_ − θ_S_ may be estimated for different values of n using one or two hypothetical datasets and computing future posterior 90% or 95% CrI’s for θ_E_ − θ_S_ and improvement probabilities Pr(θ_E_ > θ_S_ + δ|data) for δ = 0.15 or 0.20. These computations may be accompanied by an evaluation of within-arm safety monitoring rules, which are described below. In general, an SRCT should have at least n = 10 patients per arm for minimal precision, and if this is not feasible, then it probably is not worthwhile to conduct the trial.

For example, suppose that practical limitations allow at most N = 30 patients, and it is desired to study K = 3 treatments, E_1_, E_2_, and S. This implies that n = 10. If, instead, N = 45 is feasible, then n = 15, and N = 60 gives n = 20. If one can afford a total sample size up to N = 60, then for K = 3 arms, one may decide between n = 10, 15, or 20 per arm, equivalently N = 30, 45, or 60, by computing posterior results that might be obtained from hypothetical future data. Table 2 gives 95% posterior CrI’s for θ_E_ − θ_S_ and the posterior improvement probability Pr(θ_E_ > θ_S_ + 0.15|data) for different values of n and hypothetical response data on E and S, assuming non-informative beta(0.50, 0.50) priors on θ_E_ and θ_S_. In Table 2, the notation “6/10 with E vs. 3/10 with S” means that 6 out of 10 patients treated with E responded, and 3 out of 10 patients treated with S responded. Alternatively, if at most N = 30 is feasible, but per-arm sample size n = 10 is considered too small to be useful in a trial of S with two experimental treatments, E_1_ and E_2_, which are versions of a new agent given at two different doses or schedules, then as a compromise one may choose instead to conduct a two-arm trial of E_1_ and S with n = 15 per arm.

If N = 48 is feasible, and one wishes to decide between studying K = 2 or 3 treatments, then either (N, K, n) = (48, 2, 24) or (48, 3, 16). One may compare these, for example, by considering hypothetical future empirical response rates of 50% for E and 25% for S. If n = 24 and K = 2, then 12/24 responses with E and 6/24 responses with S give posterior 95% CrI [−0.02, 0.49] for θ_E_ − θ_S_, which has width = 0.51. For the smaller per-arm sample size n = 16 in a 3-arm trial of E_1_, E_2_, and S, if 8/16 responses are observed with E_1_ and 4/16 with S, this would give posterior 95% CrI [−0.08, 0.53] for θ_E1_ − θ_S_, which has width = 0.61. If, instead, one were to impose the requirement that the posterior 95% CrI for θ_E1_ − θ_S_ should have a much smaller width = 0.20, this would require n = 600 patients per arm since the empirical response rates 300/600 and 150/600 would give posterior 95% CrI [0.20, 0.30] for θ_E1_ − θ_S_ For K = 2, this would require total sample size N = 1200, making it a phase 3 trial. These examples illustrate the general fact that small samples give relatively wide CrI’s, and large samples give narrow CrI’s.

### 4.2. Constructing Within-Arm Monitoring Rules

Bayesian interim monitoring rules are very useful tools for deciding whether to drop a treatment or dose if it is found to be either unsafe or ineffective in an early-phase trial [[39],,,[46]]. To monitor θ_k_(Tox) for each E_k_ in an SRCT, one first must ask the physicians planning the trial to specify a fixed upper limit θ* on θ_k_(Tox) = Pr(Tox with E_k_) that is the largest acceptable value, and also a larger value θ** > θ* that is unacceptably large. In the numerical example given below, θ* = 0.20 and θ** = 0.40. Assuming a non-informative beta prior for θ_k_(Tox), a Bayesian criterion for stopping accrual to E_k_ is the posterior probability that θ_k_(Tox) is larger than θ*. Formally, a Bayesian posterior stopping criterion is Pr{ θ_k_(Tox) > θ*|data } > c_T_. This rule may be applied after successive cohorts of patients have been treated with E_k_ and their Tox outcomes have been evaluated. The decision cutoff c_T_, which most often is a value between 0.80 and 0.95, should be calibrated, using computer simulation of the trial, to give a small early stopping probability, P_STOP_, if θ_k_(Tox)^true^ = θ*, that is, if the probability of toxicity is acceptably low, and a large P_STOP_ if θ_k_(Tox)^true^ = θ**, that is, if the probability of toxicity is unacceptably high.

To monitor θ_k_(Res) similarly for E_k_, the physicians must specify a fixed lower limit θ* on θ_k_(Res) = Pr(Res with E_k_) that is the smallest acceptable value and also a smaller probability θ** < θ* that is an unacceptably small Pr(Res with E_k_). The posterior early stopping rule for E_k_ is then based on the posterior probability that θ_k_(Res) is smaller than θ*, given by Pr{θ_k_(Res) < θ*|data} > c_R_. In a two-arm trial of E versus S, if E is stopped early for safety or efficacy, then the trial should be stopped. In a three-arm trial of E_1_, E_2_, and S, if either E_1_ or E_2_ is stopped early, then, to improve reliability, the remaining sample of the terminated arm should be randomized among the remaining treatments. This is known as enrichment [47]. If, instead, the overall sample size is reduced after stopping accrual to an arm, this would be a false economy because it would produce a smaller precision for all final posterior estimators.

Stopping rules may be applied using several possible monitoring schedules. For example, if n = 20, then the rule may be applied at interim sample sizes of 10 and 15, while if n = 24, then it may be applied at 8 and 16. The decision cutoffs c_T_ and c_R_ may be refined [48] to change with sample size, taking the form α(n/N)^β^, where the parameters α and β are calibrated to give specific values of Pstop for θ_k_(Tox)^true^ = θ* and θ_k_(Tox)^true^ = θ**. For example, to monitor safety with n = 24 using the posterior of θ_E_(Tox), suppose that the specified upper limit is θ* = 0.20, while θ** = 0.40 is considered unacceptably high. Following Thall et al. [38,39,45], one may assume the non-informative prior θ_E_(Tox)~beta(0.20, 0.80). As a random comparator in the rule to play the role of θ* = 0.20, one may use θ_S_ (Tox)~beta(200, 800), which has a mean of 0.20 and is highly informative with ESS = 1000. If n = 24 per arm, a within-arm Bayesian safety rule may be to stop accrual to E if Pr{θ_E_(Tox) > θ_s_(Tox)|data} > c_T_. Setting c_T_ = 0.90 and applying the rule after cohorts of size 8 will stop accrual to E if.

[Number of Tox with E]/[number of patients treated with E] is greater than or equal to 4/8 or 6/16. Computer simulations show that this rule has Pstop = 0.10 for arm E if θ_E_(Tox) ^true^ = 0.20 and Pstop = 0.70 for arm E if θ_E_(Tox) ^true^ = 0.40.

### 4.3. Determining a Randomization Scheme

To facilitate safety monitoring with small samples, it is useful to restrict the randomization so that the interim per-arm sample sizes are equal each time the safety rule is applied. For example, if N = 48 patients are randomized to K = 2 arms with up to n = 24 patients each, and the within-arm monitoring rules are applied at 8 and 16 patients, then the randomization sequence may be determined so that the per-arm sample sizes are perfectly balanced at 16, 24, 32, 40, and 48 patients. To do this, one may pre-specify successive treatment assignment blocks of size 8 each, with each block a randomly scrambled sequence of four E’s and four S’s, such as (E,S,S,E, E,S,E,S). These blocks are used to assign patients to E or S as they are enrolled.

One may account for patient heterogeneity by defining subgroups (“strata”) using patient covariates that may influence clinical outcomes. Stratified randomization may then be used, with a separate randomization scheme specified within each stratum to obtain balanced sample sizes. An important caveat is that, because the per-treatment arm sample size n is small, refining this by stratification will produce very small treatment-subgroup sample sizes.

## 5. A Randomized Cell Therapy Trial

A 3-arm randomized phase 1–2 trial was conducted to compare two doses of engineered cells added to a Jak kinase inhibitor (JKI) versus the JKI alone for treating steroid-refractory acute graft versus host disease (srGVHD) in allogeneic stem cell transplant patients. Because srGVHD can be rapidly fatal [49] with a six-month survival rate of 50%, two co-primary outcomes were defined: Res = [partial, very good partial, or complete response at day 28] and Tox = [grade > = 3 regimen-related toxicity within 28 days]. An additional long-term treatment success outcome was defined as S180 = [alive without srGVHD at 180 days]. It was planned to study three treatment arms: JKI alone (Arm S), JKI + 10^6^ cells (Arm E_1_, low cell dose), and JKI + 2 × 10^6^ cells (Arm E_2_, high cell dose). A maximum of N = 48 patients would be randomized, with exactly n = 16 per arm. For example, writing θ_Ek_(T) = Pr(TOX in Arm E_k_}, if 8/16 patients responded with E_1_, and 4/16 responded with S then, assuming beta(0.50, 0.50) priors on θ_E1_(Res) and θ_S_(Res), a posterior 95% CrI for θ_E1_(Res) − θ_S_(Res) would be [−0.08, 0.53] and p{θ_E1_(Res) > θ_S_(Res) + 0.15|data} = 0.71.

The following Bayesian safety monitoring rule was used in each cell therapy arm, based on a maximum of n = 16 patients per arm. Given the fixed upper limit of 0.30 on each θ_Ek_(T) specified by the clinicians planning the trial, accrual to Arm E_k_ would be terminated early if Pr(θ_Ek_(Tox) > 0.30|data) > 0.90. This Bayesian rule formalizes the idea that the data show that the Tox rate is unacceptably high in arm E_k_. Applying the stopping rule when Tox had been evaluated for 4, 8, and 12 patients in E_k_, this posterior criterion implies that accrual to E_k_ would be stopped early if [# patients with Tox in E_k_]/[number of patients evaluated in E_k_] was greater than or equal to 3/4, 5/8, or 6/12. The randomization was restricted to balance interim per-arm sample sizes at 4 + 4 + 4 = 12, 8 + 8 + 8 = 24, and 12 + 12 + 12 = 36 to facilitate application of the safety monitoring rule. This was carried out by generating random treatment assignment sequences in 8 blocks of size 6, such as (2, 1, 3, 1, 3, 2). The operating characteristics of the within-arm safety monitoring rule are summarized in Table 3. If both cell therapy arms were stopped early, the trial would be terminated with neither E_1_ nor E_2_ selected. If one cell therapy arm was stopped early, then all patients, up to the maximum total of N = 48, would be randomized fairly between S and the remaining cell therapy arm. If neither cell therapy arm was stopped for safety, then the arm E_k_ with a larger posterior mean Pr(Res, E_k_) would be selected as best for future study.

Interim data from the cell therapy trial showed no Tox events and moderately promising efficacy for each cell therapy arm compared to S at days 28 and 180. The interim empirical response rates are summarized in Table 4a, which shows a benefit for each JKI + cellular therapy arm over JKI alone. Since the interim sample sizes were small, it is useful to do Bayesian comparisons. For the 28-day outcomes, write θ_k_(Res28) = Pr(Res28 in Arm k) for k = S, E_1_ or E_2_, and assume Beta(0.50, 0.50) priors. Bayesian posterior criteria comparing E_1_ to S and E_2_ to S in terms of posterior 95% CrI’s and probabilities of a 0.15 improvement for 28-day response and 180-day treatment success are given in Table 4b. For example, for Arm E_1_ (JKI + low cell dose), the posterior 95% CrI for the E_1_-vs-S effect, θ_E1_(Res28) − θ_s_(Res28), was [−0.05, 0.65] and Pr{θ_E1_(Res28) > θ_S_(Res28) + 0.15|data} = 0.82. Considered together, these interim results, while far from confirmatory due to the sample sizes, appeared sufficiently promising to motivate expanding the trial sample size from 48 to 96, with 32 patients for each of the three arms.

## 6. Conclusions

We have argued that small early-phase trials provide an important link between preclinical research and large confirmatory phase 3 trials and that between-treatment comparisons may be used when deciding how to proceed in the treatment evaluation process. This motivates randomizing in small trials to obtain fair treatment comparisons. We have proposed and illustrated practical Bayesian methods for comparing event rates and constructing safety and futility monitoring rules based on the data from such trials.

While randomization provides protection against biased comparisons that may result from single-arm trials, it is not a panacea. Because patients enter a clinical trial sequentially, it is not possible to balance treatment arms perfectly on patient covariates, and covariate distributions will always differ randomly between treatment arms. Additionally, while the use of safety and futility monitoring rules reduces the risk of choosing an unsafe or ineffective dose, no design is perfect, and there is always the possibility that later data will show an inference to be wrong.

Throughout most of our discussion, we implicitly have assumed homogeneity. However, as pointed out by Senn [50], it does not make sense to ignore observed patient covariates because one has randomized. Provided that a small set of prognostic covariates is prespecified in the clinical protocol, to improve precision when estimating between-arm effects defined in terms of Pr(Res) or Pr(Tox), one may fit a logistic or probit regression model including the covariates. Here, Bayesian regression is particularly useful because it does not rely on large sample approximations required by corresponding frequentist regression models. To implement these Bayesian regression models, as recommended by Gelman et al. [51], default priors may be assumed for treatment and covariate parameters. Similarly, if a substantial imbalance is seen between strata, such as males and females, then one may perform post-stratification by computing a comparative between-treatment estimate within each stratum and using these to compute an appropriately weighted average across the strata.

A final *caveat* is that while SRCTs can be very useful, it is important to resist the temptation to overinterpret positive results. While, for example, an observed response rate of 9/15 for a new treatment E_k_ may be encouraging, citing the 60% empirical response rate alone is misleading due to the small sample size n = 15. It is important to temper this optimistic estimate by quantifying one’s uncertainty. This may be carried out by giving a 95% posterior CrI for Pr(Res, E_k_), which runs from 0.35 to 0.81 for this dataset. That is, an SRCT is not a confirmatory trial.

## Figures and Tables

**Figure 1 cancers-17-01902-f001:**
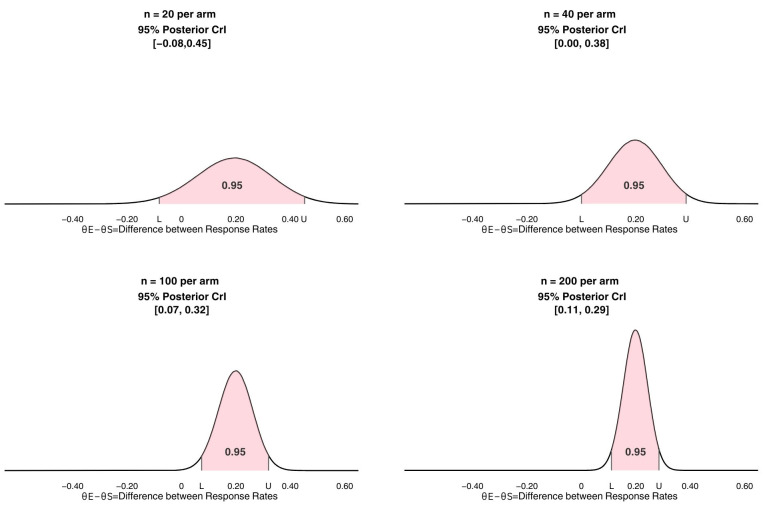
Posterior 95% credible intervals of between-treatment differences in response probabilities, θ_E_ − θ_S_, for per-arm sample sizes 20, 40, 100, and 200. For each pair of datasets, the empirical response rates are 40% for E and 20% for S.

**Figure 2 cancers-17-01902-f002:**
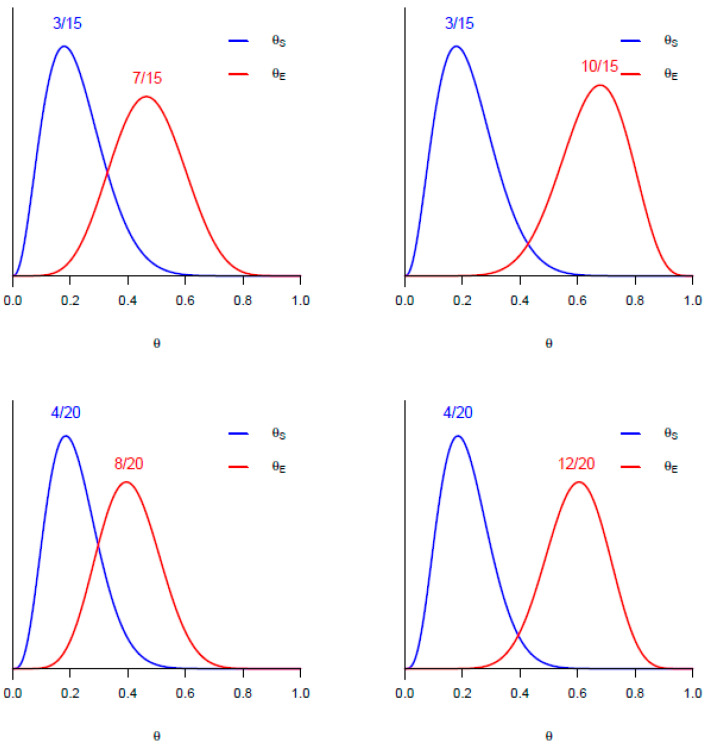
Comparisons of the posteriors of response probabilities for E and S based on data from per arm sample sizes of n = 15 (top row) or n = 20 (bottom row).

**Table 1 cancers-17-01902-t001:** Two illustrations of E-versus-S treatment comparisons, (**a**) for Good and Poor prognosis subgroups and (**b**) for biomarker Positive and Negative subgroups. True response probabilities and sample proportions are given for each treatment, E and S, in each subgroup and overall. It is assumed that Pr(Good Prognosis) = 0.20 in each sub-table, which implies that, in sub-table (**a**), Pr(Res, E) = (0.50 × 0.20) + (0.25 × 0.80) = 0.30 and Pr(Res, S) = (0.60 × 0.20) + (0.30 × 0.80) = 0.36. Similarly, in subtable (**b**), Pr(Res, E) = (0.80 × 0.20) + (0.10 × 0.80) = 0.24 and Pr(Res, S) = (0.50 × 0.20) + (0.50 × 0.80) = 0.50.

(a) Prognostic Subgroups						
Treatment	Good Prognosis		Poor Prognosis		Overall	
	True Pr(Res)	Data	True Pr(Res)	Data	True Pr(Res)	Data
E	0.50	15/30	0.25	-	0.30	15/30
S	0.60	12/20	0.30	24/80	0.36	36/100
(**b**) **Biomarker Subgroups**						
	Positive		Negative		Overall	
	True Pr(Res)	Data	True Pr(Res)	Data	True Pr(Res)	Data
E	0.80	24/30	0.10	-	0.24	24/30
S	0.50	10/20	0.50	40/80	0.50	50/100

**Table 2 cancers-17-01902-t002:** Comparisons of per-arm sample sizes n = 10, 15, and 20 in terms of posterior 95% credible intervals (CrI's) for θ_E_ − θ_S_ and posterior probabilities of at least a 0.15 improvement of E over S, Pr(θ_E_ > θ_S_ + 0.15|data).

			Posterior Quantities	
n = Number of Patients Per Arm	Overall N	Hypothetical Future Data	95% CrI for θ_E_ − θ_S_	Pr(θ_E_ > θ_S_ +0.15|data)
10	30	6/10 with E vs. 3/10 with S	−0.13, 0.63	0.74
15	45	10/15 with E vs. 5/15 with S	−0.02, 0.61	0.84
20	60	14/20 with E vs. 7/20 with S	0.04, 0.60	0.90

**Table 3 cancers-17-01902-t003:** Operating characteristics of the cell therapy trial’s within-arm safety monitoring rule.

True Pr(Tox28)	Pr(Stop Early)	Sample Size Quartiles
0.30	0.16	16, 16, 16
0.40	0.40	8, 16, 16
0.50	0.66	4, 12, 16
0.60	0.86	4, 8, 16

**Table 4 cancers-17-01902-t004:** (**a**) Observed interim treatment success rates at days 28 and 180 for the trial of JKI +/− cell therapy for steroid-refractory GVHD. (**b**) Posterior criteria for comparing E_1_ and E_2_ to S in terms of Pr(day 28 response) and Pr(day 180 success) for the trial of JKI +/− cellular therapy for steroid-refractory GVHD, computed from the data in (**a**).

(**a**)			
**Treatment Arm**		**Day 28 Res**	**Alive Without GVHD at 180 Days**
JKI		5/9 (56%)	2/6 (33%)
JKI + 10^6^ cells		9/10 (90%)	5/8 (63%)
JKI + 2 × 10^6^ cells		10/11 (91%)	7/9 (78%)
(**b**)			
**Outcome**	**Treatment Effect**	**95% Posterior Credible Interval**	**Posterior Probability of >0.15 Improvement Over S**
Res28	θ_E1_(Res28) − θ_s_(Res28)	[−0.05, 0.65]	0.82
	θ_E2_(Res28) − θ_s_(Res28)	[−0.02, 0.66]	0.83
Res180	θ_E1_(Res180) − θ_s_(Res180)	[−0.21, 0.67]	0.68
	θ_E2_(Res180) − θ_s_(Res180)	[−0.06, 0.77]	0.86

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
