# Peer review of "Practical Bayesian Guidelines for Small Randomized Oncology Trials"

_cancers, 2025, doi:10.3390/cancers17121902_

Round 1
Reviewer 1 Report
Comments and Suggestions for Authors
The article provides a concise set of guidelines that can be useful for small-sample Bayesian randomized trial design for medical treatment. This may be useful for medical practitioners who may have a somewhat limited statistics background.
One topic that I think deserves more discussion is the selection of appropriate prior models to use. As noted by the authors, the choice of prior distribution/parameters can have a significant impact on the results, particularly in the case of small samples. I think it is just as important to provide guidelines for the prior distribution selection as for the other experimental design quantities.
There are numerous wording, formatting, and grammatical issues that are minor on their own but in aggregate hinder readability. I suggest a full proofread.
For publication the authors should proofread and significantly improve the writing (I only give a subset of the grammatical issues below), and include an additional section on guidelines for selecting an appropriate prior distribution as well as parameters (in terms that can be understood and implemented by medical researchers who may have limited statistics background).
33: illustrated a -> illustrated in/by a
42: controversiaL -> controversial
47, 49: extra space
Many other occurrences of random extra spaces.
"A SRCT" -> "An SRCT"
82: What are "3+3 algorithms"?
86: triple negative is a little confusing: "not unlikely to choose an ineffective dose"
88: data show -> data shows
88-89: "data show it is completely ineffective." Missing citation?
121-122: is provide -> is to provide
Table 1: I don't see how Pr(Good prognosis) = 0.2 or Pr(Biomarker positive = 0.2) in the table as stated.
238: I don't think it is accurate to characterize the posterior value of 0.405 vs. 0.4 as being an "upward bias" "the price paid for the advantages of Bayesian estimation." It is just due to the use of the Bayesian prior, which is supposed to reflect initial beliefs and not impose bias.
Table 2: I'm a little confused about the hypothetical future data. In the first row how can it be 6/10 E and 3/10 S? That only adds to 9/10.
377/413: several periods ".."
471: default -> Default
Comments on the Quality of English LanguageOverall ok but numerous minor issues mentioned above.
Reviewer 2 Report
Comments and Suggestions for Authors
This paper builds a case for using random assignment to conditions and Bayesian estimators in early clinical assessment of new cancer treatments. The article is clear and though it will be a tough read for many who follow this journal, it does address some important points. And the use of the Bayesian metrics to compare treatments and the ideas around how to randomize and how to address toxicity in these studies seem really helpful. I have three suggestions for additions/clarifications of the text: 1) you make the case that it is well established that conventional early trials with a 3x3 design are known to make bad decisions about dosage of the experimental treatment in phase 2 trials.....it would be helpful to provide some specific examples of studies that support this conclusion and how the evidence for it has accumulated over time; 2) It would be helpful to very briefly explain the difference between Bayesian and frequentist models (most readers should know this, but the Bayesian approach is used so much less frequently and most readers would appreciate the reminder). 3) I did not see a discussion of the potential pitfalls of using this approach to design and then interrpretation of early small sample randomized trials? For example, are there examples that have used this design or interpretation approach and still got the dosage or Tox or Res wrong for Phase 2 or trials? Or have you identified any boundary conditions on these small scale randomized trials? Or is there a need to try your methods in multiple new research sequences to identify such limits?
Round 2
Reviewer 1 Report
Comments and Suggestions for Authors
Ok accept